# Blocking Studies to Evaluate Receptor-Specific Radioligand Binding in the CAM Model by PET and MR Imaging

**DOI:** 10.3390/cancers14163870

**Published:** 2022-08-10

**Authors:** Jessica Löffler, Hendrik Herrmann, Ellen Scheidhauer, Mareike Wirth, Anne Wasserloos, Christoph Solbach, Gerhard Glatting, Ambros J. Beer, Volker Rasche, Gordon Winter

**Affiliations:** 1Center for Translational Imaging, Core Facility Small Animal Imaging, Ulm University Medical Faculty, 89081 Ulm, Germany; 2Department of Nuclear Medicine, Ulm University Medical Center, 89081 Ulm, Germany; 3Department of Nuclear Medicine, Medical Radiation Physics, Ulm University Medical Center, 89081 Ulm, Germany; 4Department of Internal Medicine, Ulm University Medical Center, 89081 Ulm, Germany

**Keywords:** CAM model, HET-CAM, PSMA, chick embryo, chicken, blocking, inhibition, PET, MRI

## Abstract

**Simple Summary:**

In the development of new targeted radiopharmaceuticals, it is mandatory to demonstrate their target-specific binding. Rodents are still primarily used for these experiments. With respect to the 3Rs principles, the demand for alternative methods to reduce the number of animal experiments is continuously increasing. In the present study, we investigated whether radiotracer uptake specificity can be evaluated by blocking studies in the CAM model. PET and MR imaging were used to visualize and quantify ligand accumulation. It was demonstrated that the CAM model could be used to evaluate the target-specific binding of a radiopharmaceutical. Due to intrinsic limitations of the CAM model, animal testing will still be required at more advanced stages of compound development. Still, the CAM model could significantly reduce the number of experiments through early compound pre-selection.

**Abstract:**

Inhibition studies in small animals are the standard for evaluating the specificity of newly developed drugs, including radiopharmaceuticals. Recently, it has been reported that the tumor accumulation of radiotracers can be assessed in the chorioallantoic membrane (CAM) model with similar results to experiments in mice, such contributing to the 3Rs principles (reduction, replacement, and refinement). However, inhibition studies to prove receptor-specific binding have not yet been performed in the CAM model. Thus, in the present work, we analyzed the feasibility of inhibition studies in ovo by PET and MRI using the PSMA-specific ligand [^18^F]siPSMA-14 and the corresponding inhibitor 2-PMPA. A dose-dependent blockade of [^18^F]siPSMA-14 uptake was successfully demonstrated by pre-dosing with different inhibitor concentrations. Based on these data, we conclude that the CAM model is suitable for performing inhibition studies to detect receptor-specific binding. While in the later stages of development of novel radiopharmaceuticals, testing in rodents will still be necessary for biodistribution analysis, the CAM model is a promising alternative to mouse experiments in the early phases of compound evaluation. Thus, using the CAM model and PET and MR imaging for early pre-selection of promising radiolabeled compounds could significantly reduce the number of animal experiments.

## 1. Introduction

In vivo characterization methods are essential for developing new pharmaceuticals including targeted radiolabeled compounds for diagnostics or therapy. These especially include demonstrating that accumulation in the target tissue in vivo is indeed specific. Typically, these analyses are performed in small rodents. However, to reduce the number of test animals in the sense of the 3R principles (Refinement, Reduction, Replacement), alternative methods are in great demand. One interesting approach in this respect is using chick embryo and chorioallantoic membrane (CAM) studies, which have been known since the replacement of the Draize test by the HET-CAM model [1]. In recent years, more and more analyses have been carried out in the CAM model, as it has the potential to close a gap between in vitro studies and murine in vivo methods and thus at least has the potential to reduce the number of animals finally required [2,3,4,5,6,7,8,9,10,11]. An advantage is that no legal or ethical approvals are needed for experiments in the CAM model according to European law (Directive 2010/63/EU of the European Parliament and of the Council of 22 September 2010 on the protection of animals used for scientific purposes) as long as the animals are killed before hatching. Although the CAM model is currently not classified as an animal experiment in Germany, the specific legal situation and corresponding possible changes regarding chicken embryos must be carefully considered for each individual country. However, aside from the legal aspects, the CAM model has further advantages, making the model interesting for various applications. Within the 21 days of development of the chicken embryo, the fusion of the chorion and allantois occurs from day 5 [12,13]. This membrane is highly vascularized and, together with the natural immunodeficiency of the embryo [5,13,14,15], allows the easy establishment of tumor xenografts on the membrane. Moreover, the chicken egg model is straightforward to handle and requires only minor cost and maintenance compared to small animal models.

Based on own previous studies and data from other groups, the model appears suitable for efficient initial in vivo screening of the biodistribution of novel radiolabeled compounds prior to the evaluation in small animals [7,16,17,18,19,20,21,22,23,24,25,26]. However, testing for target specificity of a radiolabeled tracer by blocking studies has not yet been demonstrated in the CAM model. This is an essential step for evaluating the performance of novel target-specific radiolabeled compounds and is usually confirmed by using either the compound itself in unlabeled form or a competing inhibitor. In the present study, we assessed whether blocking experiments can also be performed in the CAM model with PET and MR imaging using an established prostate cancer model system [27].

## 2. Materials and Methods

### 2.1. Synthesis and Radiolabeling

Radiolabeling of the peptide [^19^F]siPSMA-14 (Technical University of Munich, Garching, Germany) was performed by isotope exchange (^18^F/^19^F). For this purpose, fluorine-18 was generated using a PETtrace 860 cyclotron (GE Healthcare, Uppsala, Sweden). Specific details of this new radiopharmaceutical are yet to be reported elsewhere. According to the manufacturer’s (Scintomics Molecular, ATT GmbH, Fürstenfeldbruck, Germany) synthesis instructions for the synthesis kit, the ligand was synthesized on a GRP cassette module. This routine synthesis achieves an average specific activity (end of synthesis) of A_S_ = (105 ± 40) MBq/µg for [^18^F]siPSMA-14.

### 2.2. Preparation of the Cell Culture

The androgen-independent and PSMA− positive (PSMA+) prostate carcinoma (PCa) cell line LNCaP C4-2 (ViroMed Laboratories, Minnetonka, MN, USA) [28] and the PSMA− negative (PSMA−) PCa control PC-3 (ACC465, DSMZ, Braunschweig, Germany) [29] were used to establish tumor xenografts in the CAM model. Based on the LNCaP C4-2 cell line, highly vascularized tumors are formed with minor hypoxia and few rim-core effects [30], while PC-3 tumors grow more invasively and proliferate strongly [31], being less vascularized and having a hypoxic and necrotic core [32]. Cell lines were cultured as described elsewhere [33]. Cell counting was performed using an improved Neubauer hemocytometer (C-chip, DHC-N01, NanoEnTek, Seoul, Korea).

### 2.3. CAM Experiments

CAM experiments were performed using a slightly modified protocol according to the previously published method [17]. Briefly, chick embryos were incubated at 37.8 °C and 65% relative humidity, starting on embryonic development day (EDD) 0. The eggshell was opened on EDD2. On EDD5, two silicone rings were placed on the CAM, and on EDD6, 0.5 × 10^6^ PC-3 (PSMA−) or 1.5 × 10^6^ LNCaP C4-2 (PSMA+) tumor cells mixed with growth matrix (30%, *v*/*v*) were applied in a total volume of 45 µL per ring. Daily monitoring of tumor growth and embryo health was performed by visual inspection. MR and PET imaging were performed on EDD15. Chick embryos were cooled at 4 °C for 120 min before MR measurement to avoid motion artifacts (according to the protocols of Bain et al. and Zuo et al. [6,34]).

For the blocking studies in the CAM model, including the associated controls, a catheter-based on a 30G needle (B. Braun, Melsungen, Germany) was placed into a blood vessel of the chorioallantoic membrane in each case. Using a catheter allows for two or more injections to be administered. Through the catheter, 100 µL of the PSMA− specific inhibitor 2-(phosphonomethyl)-pentanedioic acid (2-PMPA; Enzo Life Sciences Inc., Farmingdale, NY, USA, ALX-550-358) was injected at various concentrations (50 µM, n = 5; 0.5 µM, n = 4; 0.05 µM, n = 5; 0.005 µM, n = 5; each at 0.9% NaCl) 20 min ahead of the PET scan. Controls received either no additional application (n = 5) or an injection of 0.9% NaCl without an inhibitor (n = 3). Each respective egg was positioned together with the catheter in the PET scanner, and the application of 150 µL [^18^F]siPSMA-14 ((11.2 ± 0.3) µg/mL stock concentration) diluted in 0.9% NaCl was performed immediately after the start of the measurement. Catheter injection resulted in a higher average activity of (4.9 ± 1.0) MBq (median dose 4.6 MBq) compared to previously published experiments [17], corresponding to an average ligand concentration in ovo of (1.9 ± 1.5) µg/mL. The whole chick embryo, catheter, and syringe were measured in an activity meter (CRC-12, Capintec, NJ, USA) to determine the successfully applied radioactivity (100% injected activity [%IA]) for further quantification. A total of 33 chick embryos with tumors were selected for measurements, of which six (18%) had to be excluded due to failed injection (2), insufficient tumor growth, or large blood vessels too close to the tumor.

### 2.4. MRI and PET Measurements

For MRI, the precooled chicken eggs were placed in a custom 3D-printed holder. The holder allows MRI and PET measurements in different devices without changing the position of the egg. MR measurements were performed according to the protocols of Zuo et al. [6,35]. Data were obtained using a 60 mm quadrature volume T/R resonator on an 11.7 Tesla small-animal MRI system (Bruker BioSpec 117/16, Bruker Biospin, Ettlingen, Germany).

A T1-weighted 3D fast low-angle shot (FLASH) sequence covering the entire chicken egg was acquired as an anatomic reference for the subsequent PET ligand biodistribution measurements. The scan parameters were: TR/TE = 5/2 ms, matrix size = 400 × 400, in-plane resolution = 150 × 175 µm^2^, slice thickness = 175, no interlayer gap and NSA = 1. With 400 slices, the whole egg was covered, resulting in an acquisition time of 3 min. Furthermore, a high-resolution T2-weighted Multislice Rapid Acquisition with Relaxation Enhancement (RARE) sequence was used to accurately assess tumor volume, location, and structure. The scan parameters were: TR/TE = 4320/45 ms, matrix size = 650 × 650, in-plane resolution = 77 × 91 µm^2^, slice thickness = 500 µm, no interlayer gap, RARE factor = 8, and NSA = 4. Thirty slices were required to cover the entire tumor region, resulting in an acquisition time of 20 min.

To evaluate the biodistribution of [^18^F]siPSMA-14 in chick embryos, a dynamic 60-min scan was performed using a small-animal PET scanner (Focus120, Siemens Medical Solutions, Inc., Erlangen, Germany). The Focus120 has a high spatial resolution (<1.3 mm) and high sensitivity (approximately 7%), with a 12 cm diameter bore and 7.6 cm axial length [36]. The obtained list-mode files were processed to generate histograms (sinograms) for a time series of 23 dynamic images in frames of 6 × 20 s, 7 × 60 s, 10 × 300 s. Reconstructions were performed with OSEM3D/MAP using 4 OSEM2D, 2 OSEM3D, and 18 MAP iterations with a matrix of 256 × 256 and a zoom factor of 1.5. MRI and PET data from the chick embryos were fused by automatic rigid superposition using the PMOD software tool (PMOD Technologies, Zurich, Switzerland).

Based on the MR images, tumor xenografts of LNCaP C4-2 and PC-3 were manually selected as volume-of-interest (VOI). The placement of the VOIs is illustrated as an example in Appendix A. As part of the analysis, decay correction to the injection time was applied. For comparison, time-activity curves (TAC) of the PET data (n = 26) were generated using GraphPad Prism ver. 9.4.0 (GraphPad Software, San Diego, CA, USA). The activity concentrations for the PSMA+ and PSMA− tumor xenografts were calculated, and mean value and standard deviation (SD) were determined. As the two tumor types were each studied in the same egg, their uptake values are paired data. Therefore, the ratio of the uptakes was first formed and then averaged. The activity concentration ratios for each tumor pair (PSMA+/PSMA−) were calculated, and the mean value and standard error of the mean (SEM) were determined.

### 2.5. Statistical Evaluation

A Mann-Whitney-test and simple linear regression were performed using GraphPad Prism (ver. 9.4.0 for Windows, GraphPad Software, San Diego, CA, USA). Linear regression was conducted between 16 min p.i. and the end of the PET scan and checked for significant differences in the slopes. A *p*-value < 0.05 was assumed statistically significant.

## 3. Results

### 3.1. Tumor Size Evaluation and Visual Inspection of PET and MR Imaging

MRI was used to evaluate tumor growth in the chick embryo model. Tumor volumes were determined based on the high-resolution RARE sequence data (Figure 1).

Tumor volumes of (0.023 ± 0.011) mL for LNCaP C4-2 and (0.019 ± 0.009) mL for PC-3 were determined after eight days of tumor growth.

MRI and PET images were successfully obtained and coregistered using PMOD software, which allowed the direct correlation of the measured radioactivity to the volume of interest (VOI). These MR-based volumes correspond to an average voxel number of (4969 ± 2330) for LNCaP C4-2 and (4033 ± 1971) for PC-3. A mean activity concentration of (2.0 ± 0.9) %IA/mL was determined for the PSMA− positive tumors and (1.3 ± 0.8) %IA/mL for the PSMA− negative tumors.

For the PSMA+ tumor (LNCaP C4-2) a clear PET signal was detected without pre-dosing with the inhibitor 2-PMPA (control). Inhibitor dose-dependent differences were already apparent in the images of selected eggs (Figure 1), but quantitative analysis of all PET data was necessary for a precise evaluation. Comparing the signals of the PSMA+ and PSMA− tumor in each experiment revealed a usually lower uptake in PSMA− negative tumors (PC-3) (Figure 1).

While there was a larger difference in the PET signal between PSMA+ and PSMA− tumors at the lowest inhibitor concentration of 0.005 µM, there was almost no difference in the two highest inhibitor concentrations, 0.5 µM and 50 µM 2-PMPA, respectively (Figure 1). The signals determined in the PSMA+ tumors for the intermediate inhibitor concentration 0.05 µM start to converge, and the differences in the signals of the PSMA− tumors become less pronounced.

### 3.2. Quantitative Evaluation of PET Imaging Using Time-Activity-Curves and Linear Regression

Based on dynamic PET data and VOIs drawn in the MRI, the activity concentration [%IA/mL] for each tumor VOI was determined for the complete scan and analyzed over time (Appendix A). Time-activity curves (TACs) for each 2-PMPA-concentration experiment were generated and depicted in Figure 2 as the mean and the respective SEM.

Catheter injection allowed for the observation of early time points of the measurements. Due to perfusion effects in the tumors and neighboring blood vessels, noisy signals can be detected for the first 10 min p.i. Evaluation of the curves was therefore more reasonable starting at later time points. Linear regressions were performed for the period between 16 min p.i. and the end of the measurement. The lines were added to Figure 2, including the respective 95% confidence intervals (dotted lines).

There was no difference observable for controls without additional injection or the injection of 0.9% NaCl. Therefore, the control data were combined (Appendix A).

As expected, a significantly faster increase of [^18^F]siPSMA-14 accumulation over time was observed for the PSMA+ tumors of the control in comparison to the PSMA− xenografts, supported by a slope of (0.0182 ± 0.0005) %IA/mL/min in contrast to (0.0051 ± 0.0009) %IA/mL/min. Linear regression analyses revealed a highly significant (*p* < 0.0001) difference between the curves of the control measurements (Figure 2a).

Similar results were observed for the lowest inhibitor concentration (Figure 2b). Here, too, the radiotracer concentration increased steadily in the PSMA+ tumors (slope: (0.0112 ± 0.0012) %IA/mL/min), while only a minor increase was detected in the PSMA− tumors (slope: (0.0056 ± 0.0010) %IA/mL/min). Differences in the regression curves still are highly significant (*p* = 0.0027).

For the chicken eggs treated with a 2-PMPA concentration of 0.05 µM, also the [^18^F]siPSMA-14 concentration still increases more in the PSMA+ tumors (slope: (0.0257 ± 0.0023) %IA/mL/min) than in the corresponding PSMA− tumors (slope: (0.0133 ± 0.0026) %IA/mL/min) (Figure 2c). Although the regression curves were also significantly different (*p* = 0.0029), the differences were less pronounced than in controls or at the lowest inhibitor concentration, indicating an intermediate level of blocking.

At an inhibitor concentration of 0.5 µM, no differences in [^18^F]siPSMA-14 accumulation were detected (Figure 2d) between both tumors, PSMA+ (slope: (0.0064 ± 0.0018) %IA/mL/min) and PSMA− (slope: (0.0075 ± 0.0020) %IA/mL/min). Both TACs and linear regression lines were overlapping in the SEM and confidence intervals and the differences were not significant (*p* = 0.6831).

At the highest inhibitor concentration of 50 µM, no differences were detected between the two tumor types (Figure 2e). The increase in [^18^F]siPSMA-14 concentration in the PSMA+ tumor (slope: (0.0151 ± 0.0014) %IA/mL/min) was nearly identical to the PSMA− tumor (slope: (0.0152 ± 0.0008) %IA/mL/min). A similar trend was demonstrated for both TACs, and no significant difference was observed (*p* = 0.9916).

The clear difference in TACs between the PSMA+ and the PSMA− tumors of the controls already indicated a PSMA− specific accumulation of [^18^F]siPSMA-14 (Figure 2a). This observation was supported by the results that radiotracer accumulation could be blocked by the administration of increasing concentrations of the PSMA− specific inhibitor 2-PMPA in the PSMA+ tumors (Figure 2b–e). Accordingly, these experiments also support the hypothesis that the CAM model is a potentially suitable candidate for inhibition studies to assess receptor-specific accumulation.

### 3.3. Analysis of Ratios between the PSMA+ and the PSMA− Tumors

The activity concentration values of the last frame of each chicken egg measurement were used to determine the ratios between PSMA+ and PSMA− tumors given as mean ± SEM (Table 1).

Without pre-dosing, a significantly higher accumulation in the PSMA+ tumor was observed based on the PET data for the control, as indicated by the PSMA+/PSMA− ratio of 2.57 ± 0.60 (Figure 3). Similar results were observed for the lowest inhibitor concentration. Injecting 0.005 µM of 2-PMPA, a PSMA+/PSMA− ratio of 2.48 ± 0.27 for the PET measurement was obtained. These results indicated insufficient inhibitor concentration to block the uptake of [^18^F]siPSMA-14.

Application of an inhibitor concentration of 0.05 µM slightly reduced uptake in PSMA+ tumors, and the activity ratio of PSMA+/PSMA− decreased to 1.64 ± 0.31 (Figure 3).

Beyond an inhibitor concentration of 0.5 µM, the measured activity concentrations were balanced, resulting in PSMA+/PSMA− ratios of 0.89 ± 0.10. Accordingly, at the highest inhibitor concentration used, 50 µM, a ratio of activity concentrations close to 1 was determined (1.21 ± 0.04), indicating identical and nonspecific activity concentrations in both tumor models.

A clear inhibitor-dependent trend in tumor accumulation was demonstrated, based on the ratio evaluations. While no inhibition occurred at a low concentration, as expected, partial inhibition was achieved by the administration of 0.05 µM 2-PMPA, and complete inhibition was achieved at high concentrations of 0.5 µM and 50 µM.

## 4. Discussion

We successfully demonstrated, exemplified by using [^18^F]siPSMA-14, that target-specific tumor accumulation of a radiotracer can be assessed by inhibition studies in the CAM model with combined PET and MR imaging. In the model, various levels of inhibition could be detected, corresponding to the applied inhibitor concentration, demonstrating the potential of the method for quantifying receptor occupancy by a given target-specific radiopharmaceutical.

Due to the limitations of the CAM model in terms of biodistribution and pharmacokinetics, additional animal studies will always be needed at more advanced stages of compound development; however the model has a high potential for the early stages of characterization of new compounds. This will enable the pre-selection of promising compounds, thus reducing the number of animal studies required in concordance with the 3Rs principles.

### 4.1. Static Evaluation of PET-Data

We successfully demonstrated the specific blocking of radiotracer accumulation in the CAM model by PET and MR imaging. In the case of a complete inhibition, 2-PMPA occupies all free PSMA binding sites and accumulation in both tumors is characterized by nonspecific mechanisms, e.g., general perfusion and the EPR effect due to possible co-transport with albumin. Thus, the PSMA+/PSMA− ratio should be at a value of 1, as the nonspecific accumulation should be equal in both the blocked PSMA+ and PSMA− tumors, as was demonstrated in our experiments. The results of this study indicate complete inhibition both at the two highest concentrations used (50 µM and 0.5 µM). Both ratios are close to 1 and differ significantly from the PSMA+/PSMA− ratios of the control measurements, which reflects a specific accumulation of the tracer [^18^F]siPSMA-14. Almost identical ratios for the control and the PSMA+ tumors blocked with the lowest used concentration of 0.005 µM 2-PMPA suggests no substantial inhibition of radiotracer accumulation at this low concentration. Furthermore, partial inhibition was observed at the intermediate concentration of 0.05 µM with a PSMA+/PSMA− ratio at a level between the complete and no inhibition results, which suggests that not all binding sites were saturated at this intermediate concentration.

Our data concerning dose-dependent inhibition were in excellent agreement with the expectations based on the K_i_ value of 0.275 nM for the inhibitor 2-PMPA reported in the literature [37,38]. The small molecule 2-PMPA is a common inhibitor of PSMA and is regularly used for inhibition studies [39,40,41,42,43,44,45,46,47,48,49,50,51,52], with no side effects described at the concentrations used in this study. The affinity of the compound is about two orders of magnitude higher compared to [^18^F]siPSMA-14, with an inhibitory concentration 50% (IC50) of (13.0 ± 1.2) nM based on patent information [53], and thus was a suitable inhibitor for the present study. Calculations of 2-PMPA concentrations in chicken embryos revealed inhibitor concentrations in the chicken embryo of 0.01 nM, 0.1 nM, 1 nM, and 109 nM at an average volume of 46.5 ± 3.2 mL. Thus, while the lowest concentration used in our experiments was below the K_i_, the two highest concentrations used were one to four orders of magnitude higher than the published K_i_ and the intermediate concentration used was of the same order of magnitude as the reported K_i_.

Due to the experimental approach, the concentration of the applied ligand was in some experiments higher than the concentration of the inhibitor. An excessive amount of cold peptide could have been the cause for a low PSMA+/PSMA− ratio, especially in the controls or the lower inhibitor concentrations. Despite the higher ligand concentration, specificity was demonstrated by the inhibition studies, as the PSMA affinity of the inhibitor was significantly higher than of the ligand. However, the ligand concentration should be considered in similar experiments to ensure optimal differences between ligand and inhibitor.

Calculation of individual ratios of tracer accumulation between PSMA+ and PSMA− tumors yielded robust values for the analyses. The activity concentrations for the same tumors could vary between individual experiments in one group. However, since these fluctuations affect both tumor xenografts equally in the particular egg, the ratios could be used to compensate for these differences.

### 4.2. Dynamic Evaluation of PET-Data

Evaluation of the time-activity curves using linear regression confirmed the results of the static ratio analyses. While the slopes of the accumulation kinetics in the controls for PSMA+ and PSMA− tumors were significantly different, no significant difference between the kinetics of tumor accumulation could be detected for blocking with 50 µM. Thus, the kinetics of radiotracer accumulation in the PSMA+ tumors were consistent with the kinetics for nonspecific uptake in the PSMA− tumors. Even at a concentration of 0.5 µM 2-PMPA, complete inhibition could be demonstrated based on the kinetics, as no significant difference between the tumors could be detected either.

Significant differences in accumulation kinetics were detected for both controls and the lowest concentration of 2-PMPA used (0.005 µM). Here, the linear regression analyses also confirmed the evaluation via the ratio calculations, and we could demonstrate conclusively that there was no substantial inhibition of radiotracer uptake at this concentration.

At the intermediate concentration no complete inhibition was observed. The time-activity curves and the linear regression evaluations also implied a dose-dependent incomplete inhibition. The results were not as distinct as for the ratio calculations, since both activity concentrations over time, for PSMA+ and PSMA−, were relatively high in these experiments.

While the time-activity curves already provided a good indication of the different accumulation kinetics in the tumor xenografts, the linear regression analysis facilitated a quantitative evaluation and assessment of the statistical significance of the differences. As expected, due to perfusion effects, the data obtained during the first 10 min of the measurements is often more noisy. These effects can be minimized by starting the linear regression analysis at a later time point, thus reducing signal variability.

The time-activity curves show the accumulation of the applied substance in a selected region over time. The use of relative activity concentration should normalize the data for comparability. However, kinetics also depend on the input function, including factors such as the concentration of the substance, the injection rate, or the specifics of the animal model [54,55,56,57,58]. It is therefore difficult to compare the curves of the different concentrations with each other. We suspect that such effects have a greater impact on the evaluation in the CAM model than in the murine model due to the smaller tumor structures and associated partial volume effect (PVE).

### 4.3. Limitations

The CAM model has intrinsic limitations, as we extensively discussed in a previously published study [17]. In the present study, the small size of the tumor xenografts was a particular challenge. Small anatomical structures smaller than three times the full width at half maximum (FWHM) are affected by the PVE. Consequently, in the case of Focus 120, structures 3.39 mm and smaller are affected [16,17,36], which includes the smaller tumors in the CAM model. The selection of ^18^F as radionuclide avoids resolution degradation due to positron range, as this is 0.6 mm for ^18^F, below the optimal spatial resolution (1.13 mm FWHM; tangential, filtered back projection) of the Focus 120 PET scanner used. While partial volume correction in chicken eggs still requires considerable effort, a PVE factor of small animal imaging could be optimized with less effort. We are working on a method to either compensate for the resolution effects using PVE correction or incorporating a PVE factor for small animal imaging and plan to provide a solution to this problem in future studies.

Also, γ-counter measurements are usually considered the gold standard for quantifying the accumulation of radiolabeled substances. Accurate extraction of small structures, such as CAM xenografts, for gamma counter measurements, including separation of undesired tissue and blood debris, can be difficult and lead to erroneous measurement results. Therefore, we focused on PET evaluation in this study.

For inhibition studies, but also for binding studies, we recommend using different tumor models that are either positive or negative with respect to the expression of the corresponding target structure, as was done in our studies. Immunohistochemical analyses can then be used to detect the expression in the tumor entities and, if necessary, to determine the ratio of the expression levels between the different tumors. For PSMA, we have already demonstrated in a recent publication that no PSMA expression was observed for PC-3 [17].

Radiotracer injection is often challenging in the CAM model [17]. Due to the small and sometimes poorly accessible blood vessels in the CAM, injecting once, let alone twice, is challenging. Thus, the injection method was optimized, and two reliable consecutive injections were done using a simple catheter. Administration of 0.9% NaCl 20 min prior to the PET study did not show any significant changes in tumor accumulation. Consequently, changes in tumor accumulation in the blocking studies were attributed to the inhibitor and were not to the application method or solvent. Obtained results for the controls were also in good agreement with previous data [17].

First descriptions of successful catheterizations of blood vessels in CAM, as well as first pharmacokinetic measurements, have already been published. The methods used for catheter placement, either by microsurgery or by injection through the shell membrane during candling, are very well described [21,59]. In addition to a comprehensive review on the CAM model [60], Chen et al. also recently published a paper in which relevant experiments based on catheterization were performed [61]. The different approaches to the same method provide an excellent basis for using the CAM model in conjunction with catheter application.

For future studies, catheterization of blood vessels in the CAM model enables the biodistribution analysis starting with the injection.

### 4.4. Perspective

In our previous publications, we demonstrated the accumulation of ^68^Ga- or ^18^F-labeled PSMA ligand in the respective xenografts in the CAM model. Preliminary indications of the possibility to analyze pharmacokinetics and the general description of the methodology in the CAM model were included in these, as well as indications of PSMA expression based on histological and immunohistochemical analyses [16,17]. However, these studies lacked evidence on the specificity of this accumulation.

Specificity is usually demonstrated by blocking studies using either the compound itself in unlabeled form or a competing inhibitor. The blocking agent is administered in excess through a separate injection by pre-dosing 10 min to 30 min prior to compound application [48,49,50,51,52] or by co-injection with the substance to be analyzed [39,40,41,42,43,44,45,46,47]. Due to the lack of information on the pharmacokinetics of the inhibitor in the CAM model so far and also to demonstrate the feasibility of multiple catheter applications in this study, we opted for preinjection of the inhibitor 20 min prior to the radiolabeled compound. For future studies, co-injection of the inhibitor and compound can be tested to evaluate whether inhibition studies can be performed on a single injection basis.

Another critical variant in inhibition studies is the analysis of specific binding both with and without an inhibitor in the same animal on consecutive days. A prerequisite for this measurement is an appropriately short-lived radionuclide, which would be given, for example, with ^68^Ga or ^18^F. Furthermore, it should be ensured that the ligand from the first application, either due to the low concentration used or also due to appropriate pharmacokinetics and excretion, has no influence on the binding in the second application. If these conditions are met, this form of measurement will allow the detection of binding and inhibition in the same animal and thus additionally reduce the number of animals required in terms of the 3Rs principles. In the CAM model, repeated measurements with imaging modalities may put too much stress on the embryo. In this case, the cooling or anesthesia necessary to immobilize the embryo may be a limiting factor for the embryo’s survival.

Additionally, repeated application in the chicken egg on different days can be challenging. If the catheter remains in the egg until the second injection, the longer residence times in the chicken model can potentially result in increased lethality, e.g., due to injuries with the needle caused by movements of the embryo. Placement of a new catheter the next day may be problematic due to the availability of suited blood vessels. We are already conducting studies with multiple measurements in the context of the CAM model to evaluate the potential for follow-up studies over various days, but so far, only with single injections. The described evidence of binding and specificity on consecutive days of measurement in the same animal will be analyzed in future studies to further test the capabilities of the CAM model.

In the development of new radiopharmaceuticals, the demonstration of specific binding is an essential step to clinical application. The analysis in the murine model for each new ligand causes a high demand for animal experiments, corresponding animal test applications and the associated effort and costs. The CAM model allows initial binding and specificity studies, thus narrowing down the candidates for the murine assays. Experiments can be performed rapidly, provided that no animal test application is required for the CAM model, as is currently the case in many countries. Even if an animal experiment application is required, the lower cost, reduced preparation time, and simpler housing conditions are advantages of the CAM model over the murine standard. While there is no doubt that the CAM model cannot completely replace murine models, the pre-selection of the appropriate candidate ligand from an often large number of potential structures can be accelerated by using the CAM model. It also helps to decrease of the number of experiments required in mouse models.

## 5. Conclusions

The present results support our hypothesis that the CAM model offers great potential to reduce the required number of initial animal experiments during the development of new radiopharmaceuticals. Concerning the detection of specific drug accumulation employing inhibition or pre-dosing studies, the CAM model can be considered as an alternative to, e.g., animal experiments in mice. From our point of view, further studies are necessary to explore the potential of the CAM model for pharmacokinetic analyses. Since the chick embryo is significantly different from adult small animals, additional animal experiments for the final radiopharmaceutical characterization cannot be completely replaced.

## Figures and Tables

**Figure 1 cancers-14-03870-f001:**
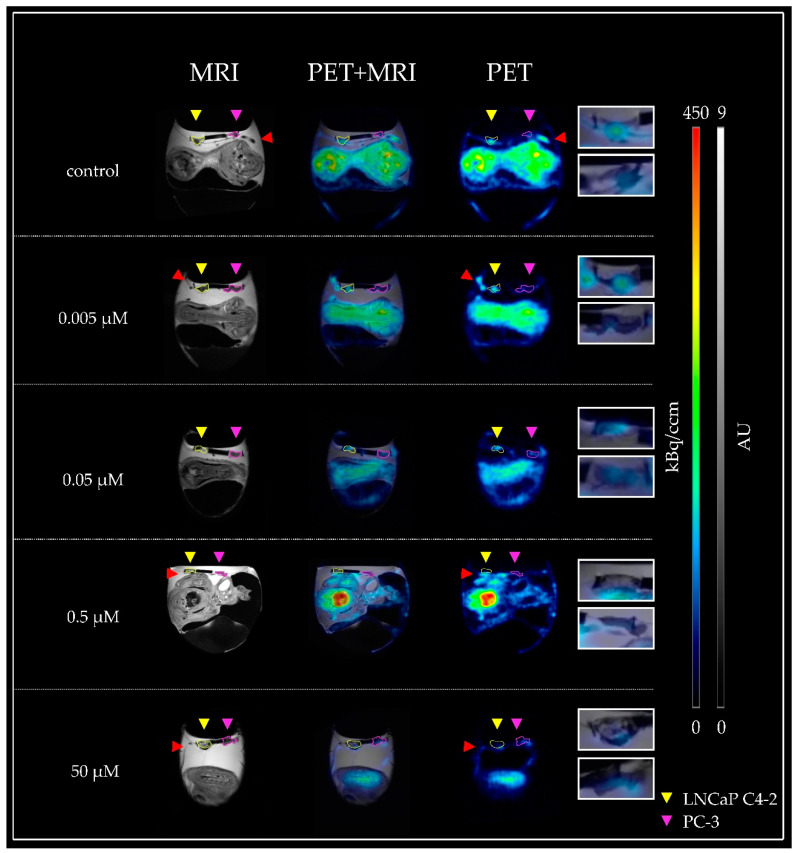
Representative MR and PET images of the CAM model injected with various inhibitor concentrations. From top to bottom, the applied inhibitor concentration was increased. Tumor regions were obtained by a T2-weighted RARE scan (**left**), while a static reconstruction of a 60 min PET scan was used to demonstrate the accumulation of the radioligand (**right**). In the resulting fusion image (middle), a marked accumulation of [^18^F]siPSMA-14 could be localized in the PSMA− positive tumor LNCaP C4-2 (yellow arrow, left tumor) for the control and the lowest inhibitor concentration (0.005 µM). While [^18^F]siPSMA-14 accumulation in LNCaP C4-2 tumors decreased with increasing inhibitor concentration, [^18^F]siPSMA-14 accumulation in PC-3 (magenta arrow, right tumor) remained approximately the same for all concentrations. For better visualization, the tumor xenografts were delineated in color. Magnified images of the corresponding tumor regions are depicted on the right in separate boxes (LNCaP C4-2 upper box, PC-3 lower box). The signal in the blood vessels (red arrows) must be carefully distinguished from accumulation in the tumor.

**Figure 2 cancers-14-03870-f002:**
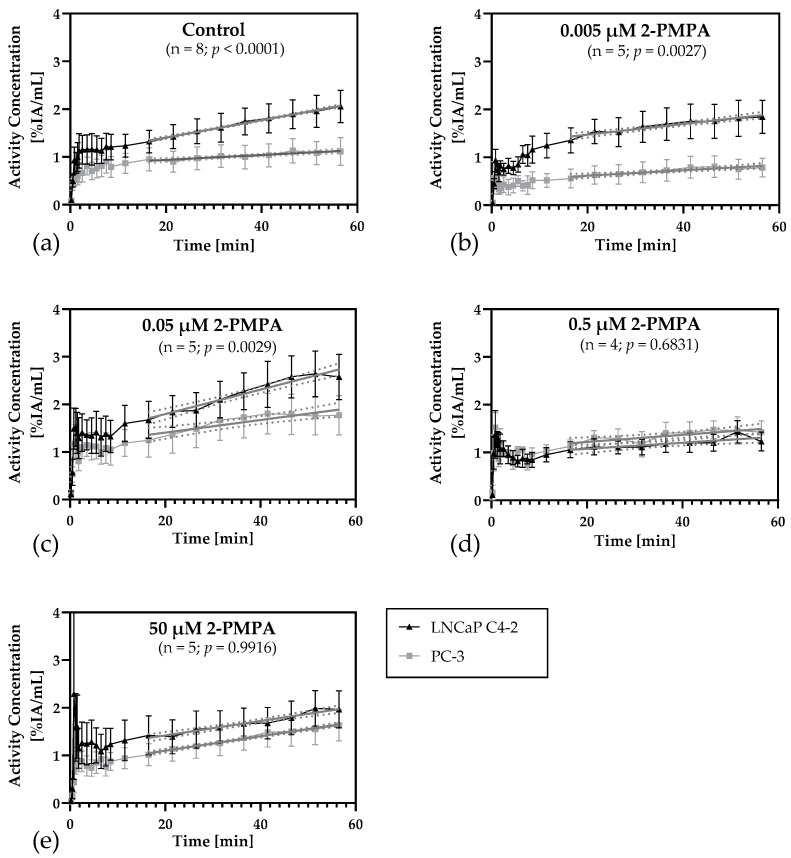
Time-activity curves of [^18^F]siPSMA-14 accumulation of control measurements and various inhibitor concentrations based on the mean and SEM of the respective experiments. Additionally, the linear regression lines and their respective 95% confidence intervals (dotted lines) starting 16 min after scan initiation have been added. A *p*-value <0.05 indicates significant differences in the slopes of the graphs based on linear regression analysis. For all inhibitor concentrations (**b**–**e**) and including the control (**a**), only a minimal increase in radiotracer activity concentration [%IA/mL] is observed for the PSMA− negative tumors, as indicated by the regression lines. While at the lowest inhibitor concentration (**b**) and in the control (**a**) there was a significantly higher activity concentration detectable in the PSMA+ tumors, and differences in the [^18^F]siPSMA-14 uptake between the two tumor types decreased with increasing inhibitor concentration. At an inhibitor concentration of at least 0.5 µM 2-PMPA there are no significant differences between PSMA+ and PSMA− tumors, as indicated by the overlapping SEMs. This suggests a specific inhibition of [^18^F]siPSMA-14 in PSMA+ tumors with increasing 2-PMPA concentration.

**Figure 3 cancers-14-03870-f003:**
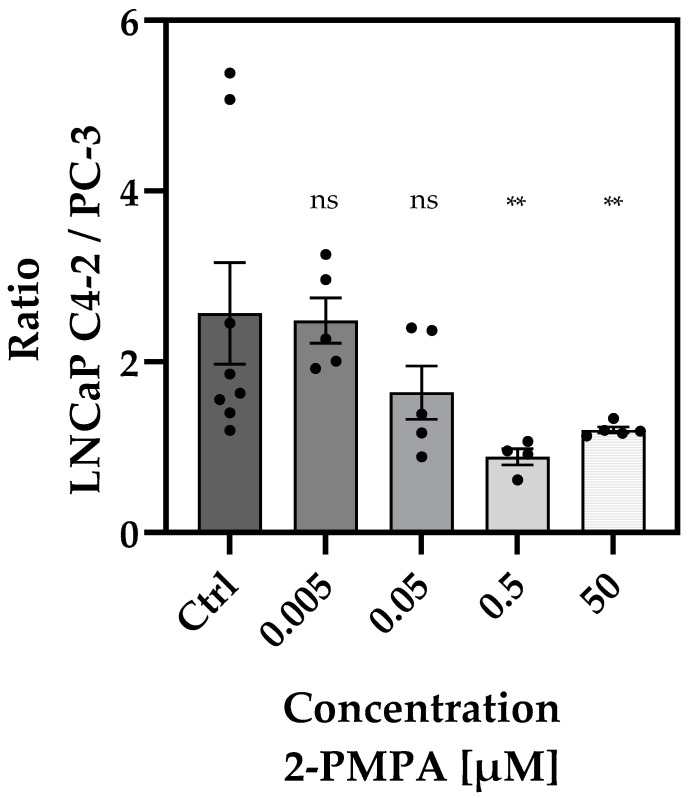
Overview of the ratios of [^18^F]siPSMA-14 uptake in PSMA+ vs. PSMA− tumors treated with various 2-PMPA concentrations. For untreated control (Ctrl), a ratio of 2.6 ± 0.6 was determined. No significant difference to the Ctrl was observed for the lowest inhibitor concentration of 0.005 µM and also for 0.05 µM. A significant difference was observed for an inhibitor concentration of 0.5 µM, indicating an equal tracer uptake in PSMA+ and PSMA−. A similar ratio was determined for the highest inhibitor concentration of 50 µM 2-PMPA. Distinct values are summarized in Table 1. Each circle represents a single experiment. Concentration ratios between the control and the various concentrations were assumed to be significantly different for *p* < 0.005 in the Mann-Whitney test. ns = not significant; ** *p* < 0.005.

**Table 1 cancers-14-03870-t001:** PSMA+/PSMA− ratios of activity concentrations for the respective 2-PMPA concentration. The values are given as the average of the ratios and the respective SEM.

2-PMPA Concentration	PSMA+/PSMA−
*w*/*o*	2.57 ± 0.60
0.005 µM	2.48 ± 0.27
0.05 µM	1.64 ± 0.31
0.5 µM	0.89 ± 0.10
50 µM	1.21 ± 0.04

## Data Availability

The used data, additional to those in the Appendix A, are available from the corresponding author on reasonable request.

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
