# Peer review of "Blocking Studies to Evaluate Receptor-Specific Radioligand Binding in the CAM Model by PET and MR Imaging"

_cancers, 2022, doi:10.3390/cancers14163870_

Round 1

Reviewer 1 Report

The authors have taken the comments into account and the amended manuscript is sound and of interest to the scientific community.

I would just like to refer to my first comment about tumour volumes and the request to provide the number of voxels for PET VOIs.

Indeed, tumours look very small on MR and PET images and my question was about the low resolution of PET compared to MRI and PET data analysis performed on such small volumes. In the revised manuscript, the authors have now given the mean voxel number, which seems to be very high for PET data with the mean volume given. Could the authors please check whether the numbers they have included are the correct ones for the PET VOIs, or whether these voxel numbers refer to the MRI data?

Reviewer 2 Report

The authors addressed any concerns from the initial review in great detail. From my perspective, the manuscript is ready for publication.

Reviewer 3 Report

The authors have fully addressed my comments. I have not further comment!

This manuscript is a resubmission of an earlier submission. The following is a list of the peer review reports and author responses from that submission.

Round 1

Reviewer 1 Report

Manuscript: cancers-1787646:

Content:
The manuscript “
Blocking Studies to Evaluate Receptor-Specific Radioligand Binding in the CAM Model by PET and MR Imaging” by Löffler et al is an original article describing the use of the chick chorioallantoic membrane assay as an alternative approach in PET radiotracer development and evaluation, in particular regarding aspects of radiotracer specificity. This approach is 100% in line with the 3R principles, here in particular the replacement and reduction of animal experiments, a valuable and absolutely supportive initiative.

In order to test the feasibility of investigating radiotracer uptake specificity the authors performed PET and MR imaging using the PSMA-specific ligand [18F]siPSMA-1 with and without pre-blocking by the corresponding PSMA inhibitor 2-PMPA. This work is important and has a high impact for radiotracer development, however some aspects need to be clarified.

Results part 3.1 & 3.2

1.      Tumours on PET and even MR images are not easy to spot. I appreciate the magnified images on LNCaP C4-2 tumours (Figure 1), but magnified images of both PSMA-positive and negative tumours with clear delineation of tumour would facilitate observations on PET AND MR images. The way Figure 1 is presented, I am not able to clearly identify the tumours.

Furthermore, regarding tumour size (indicated in mL), could the authors indicate how many voxels this reflects for the PET VOIs? Also, here, could there be an example in supplementary data showing tumor VOI definition on MRI and corresponding PET images?

Page 6, line 3 please indicate mean +/-SD %ID/mL for PSMA+ and PSMA- tumors.

2.       An average activity of 4.9 +/- 1.0 MBq was injected for the CAM model. This sounds relatively high, in particular as this is more than the double of what was used in the previous study (Löffler et al 2021, Cancers; 2.4 +/- 0.9 MBq and 3.3 +/- 1.6 MBq were used for the CAM and mouse model, respectively). Could you explain?

3.       For blocking studies, the authors used the PSMA inhibitor 2-PMPA and not unlabeled si-PSMA-14. Please explain why and discuss if there are differences in binding affinities between the two compounds.

What is the rationale of the 20-min blocking time slot before PET scan, have there been any calculations for equilibrium related to that time point?

4.       Figure 2 and Table 1, there is some inconsistency for the blocking results. With increasing concentration of blocking agent, there is no continuous decrease in radiotracer uptake, which can be clearly seen on the TACs and is also reflected in the slope values. One can find a decrease in slope with the first concentration of 0.005µM 2-PMPA from 0.0182+/-0.0005 to 0.0112+/- 0.0056 %IA/mL/min for the PSMA+ tumour. However, when using 0.05µM 2-PMPA the slope increases to a value of 0.0257+/-0.0023 %IA/mL/min, which is even higher as without 2-PMPA. Furthermore, at the highest blocking concentration of 50µM 2-PMPA, slope values for both PSMA-positive and negative tumours are equally high with 0.0151+/-0.0014 or 0.0152+/-0.0008 %IA/mL/min, which are close to the value without blocking for the PSMA-positive tumor. I have difficulties to understand this and in the discussion part this point is not addressed.

Discussion part 4.2

5.       Page 11 line 1-3: In line with my previous comment, the authors mention that "the calculation of the slope ratios of the linear regression" shows an "evident dose-dependent inhibition of tumour accumulation", pointing at the following values (Ctrl=3.6; 0.05 µM=1.9; 50 µM=1.0).

If I calculate the missing values (based on the data from table 1), 0.005 µM=10.18 and 0.5 µM=0.85), this is not as evident as mentioned. There should be some explanation or at least discussion.

Reviewer 2 Report

The authors have undertaken an important validation of CAM models for evaluating PET radiotracers. The experiments were well designed and explained in the paper. There were a couple concerns to address before publication:

1) in Figure 1, it is difficult to visualize the differences between blocking and baseline images due to separate subjects in each experiments. Are longitudinal studies in the same subject possible with the CAM model? In particular, day 1 imaging at baseline and day 2 with blocking? To be clear, this is not a request to complete these experiments, rather if a line to explain something to this effect could be added to the discussion it would help readers better understand the utility of CAM models.

2) There are several mentions of the lack of required ethical approval to use the CAM model. This is risky as regulations are constantly changing, often leaning to more stringent. As it occupies the space between in vitro and in vivo, ethics of using in ovo may also have differing regional or cultural opinions. I caution the authors of over relying on this as an advantage of the CAM model.

3) Does the PSMA inhibitor used in the blocking experiments activate any physiological systems? For instance, does it change perfusion, potentially altering the uptake in those experiments. The non-specific uptake increases in the higher concentrations of the blocking experiments, which could be more available PSMA tracer or changed delivery. 

4) the specific activity for the 18F-Si-PSMA-14 was low. MW=1473, egg volume = 46.5 mL, As=105 MBq/ug = 11.2 ug/mL injected x 150 uL ==> I caluclated 24.5 nM concentration of 19F-Si-PSMA-14. This is higher than some of the blocking studies. May explain why the S:NS binding is so low in the baseline experiments between the PSMA+/PSMA- tumors. This was also observed in the Cancers 2021 paper from the same group between CAM and mouse models.

5) immunohistochemical analysis of the tumors to quantify the PSMA expression to understand what the expected ratio between LNCaP and PC-3 tumors. Don't need those numbers here but a mention of this as a limitation to this study would help anyone planning on using CAM models to appropriately plan their own experiments.

6) is there any metabolism of the tracers in ovo? Not sure if it would impact the results of this experiment but could be an additional advantage of using the CAM model if there is less metabolism to confound preliminary binding and blocking experiments.

Overall it was an excellent paper on an alternative to rodent models for radiotracer screening. Innovative approaches such as this are needed for the field to keep up with growing demand for targeted PET probes.

Reviewer 3 Report

The manuscript entitled " Blocking Studies to Evaluate Receptor-Specific Radioligand Binding in the CAM Model by PET and MR Imaging" by Winter* et al., has proposed a study on evaluating whether radiotracer uptake specificity can be evaluated by blocking studies in the CAM model. Overall, the manuscript is well organized and written, and the data are fruitful and convincing enough to support the point of view. Therefore, I suggest this manuscript be accepted by Cancers.

Reviewer 4 Report

The current study by Jessica Löffler et al. shows that CAM can be an alternative for in vivo binding capabilities of radiopharmaceuticals to reduce animal use by multimodal PET and MR imaging. I have the following comments.

·       Development of feasible and translation in vivo model for radiopharmaceuticals is of high priority in development of novel radioligands. Dedicated for this purpose, the authors present a CAM model, based on prostate cell lines (both PSMA+ and PSMA-), with excellent quantification analyses from multimodal imaging techniques.

·       The following studies (PMID: 33671534, PMID: 23970367) are highly relevant to the topic you studied. Please elaborate on this in the discussion section.

·       Confirmation of the results here is suggested with additional cell line as a routine practise.

·       The author team have a foundational publication on this topic (PMID: 32429233), it is suggested to elaborate on continuity between these two studies to provide more deepness to readers.

·       There are some typos in throughout the manuscript, for instance, “in ovo” in the abstract section. I assumed it would be “in vivo”. Please double check the manuscript.

·       Finally, the study is methodologically sound and performed with high precision, from CAM model to multimodal imaging, which guarantee the quantifiable measures. No major flaw was found in the study.

Reviewer 5 Report

I am very pelased to review the study submitted by Jessica Löffler and co-authors.

The study showed brilliant results demostrating that [18F]siPSMA-14 that target-specific tumor accumulation of a radiotracer can be assessed by inhibition studies in the CAM model by combined PET and MR imaging.

I have only a suggestion. Despite the nature of the study is focused on a pre-clinical animal design, the authors should briefly discuss the future prospective of these results in translational research and potential development in clinical setting.
